# REVISITING THE GENERALIZATION OF ADAPTIVE GRADIENT METHODS

## ABSTRACT

A common belief in the machine learning community is that using adaptive gradient methods hurts generalization. We re-examine this belief both theoretically and experimentally, in light of insights and trends from recent years. We revisit some previous oft-cited experiments and theoretical accounts in more depth, and provide a new set of experiments in larger-scale, state-of-the-art settings. We conclude that with proper tuning, the improved training performance of adaptive optimizers does not in general carry an overfitting penalty, especially in contemporary deep learning. Finally, we synthesize a "user's guide" to adaptive optimizers, including some proposed modifications to AdaGrad to mitigate some of its empirical shortcomings.

## 1 INTRODUCTION

Adaptive gradient methods have remained a cornerstone of optimization for deep learning. They revolve around a simple idea: scale the step sizes according to the observed gradients along the execution. It is generally believed that these methods enjoy accelerated optimization, and are more robust to hyperparameter choices. For these reasons, adaptive optimizers have been applied across diverse architectures and domains.

However, in recent years, there has been renewed scrutiny on the distinction between adaptive methods and "vanilla" stochastic gradient descent (SGD). Namely, several lines of work have purported that SGD, while often slower to converge, finds solutions that generalize better: for the same optimization error (training error), adaptive gradient methods will produce models with a higher statistical error (holdout validation error). This claim, which can be shown to be true in a convex overparameterized examples, has perhaps muddled the consensus between academic research and practitioners pushing the empirical state of the art. For the latter group, adaptive gradient methods have largely endured this criticism, and remain an invaluable instrument in the deep learning toolbox.

In this work, we revisit the generalization performance of adaptive gradient methods from an empirical perspective, and examine several often-overlooked factors which can have a significant effect on the optimization trajectory. Addressing these factors, which *does not require trying yet another new optimizer*, can often account for what appear to be performance gaps between adaptive methods and SGD. Our experiments suggest that adaptive gradient methods do not necessarily incur a generalization penalty: if an experiment indicates as such, there are a number of potential confounding factors and simple fixes. We complete the paper with a discussion of inconsistent evidence for the generalization penalty of adaptive methods, from both experimental and theoretical viewpoints.

### 1.1 OUR CONTRIBUTIONS

Our work investigates generalization of adaptive gradient methods, and constructively comments on the following:

**The brittleness of simple experiments and simple abstractions.** We attempt a replication of the experiments from Wilson et al. (2017), finding that they have not stood up to unknown hardware and software differences. We show simple theoretical settings where adaptive methods can either

fail or succeed dramatically, as compared to SGD. Though each can shed interesting insights, neither abstraction is reflective of the truth.

**The perils of choosing a large $\epsilon$.**  The innocuous *initial accumulator value* hyperparameter destroys adaptivity at parameter scales smaller than $\sqrt{\epsilon}$. This really matters in large-scale NLP; a foolproof solution is to use our proposed "$\varepsilon = 0$" variant of AdaGrad.

**The subtleties in conducting a proper optimizer search.**  The differences between Adam, Ada-Grad, and RMSprop are not fundamental; some, like AdaGrad's lack of momentum, are easily fixable. Upon disentangling these differences, and with enough tuning of the learning rate schedule, we suggest that all three are equally good candidates in optimizer search, and can match or beat SGD.

## 1.2  RELATED WORK

Adaptive regularization was introduced along the AdaGrad algorithm in parallel in (Duchi et al., 2011; McMahan & Streeter, 2010). A flurry of extensions, heuristics and modifications followed, most notably RMSprop (Tieleman & Hinton, 2012) and Adam (Kingma & Ba, 2014). Today, these papers have been cited tens of thousands of times, and the algorithms they propose appear in every deep learning framework. For an in-depth survey of the theory of adaptive regularization and its roots in online learning, see (Hazan, 2016).

Upon a quick perusal of recent literature, there is plenty of evidence that adaptive methods continue to be relevant in the state of the art. Adam in particular remains a staple in recent developments in fields such as NLP (Devlin et al., 2018; Yang et al., 2019; Liu et al., 2019), deep generative modeling (Karras et al., 2017; Brock et al., 2018; Kingma & Dhariwal, 2018), and deep reinforcement learning (Haarnoja et al., 2018). Adaptive methods have seen adoption in extremely large-scale settings, necessitating modifications to reduce memory consumption (Shazeer & Stern, 2018; Anil et al., 2019; Chen et al., 2019).

In recent years, there have been various works attempting to quantify the generalization properties of SGD. These varied perspectives include general analyses based on stability and early stopping (Hardt et al., 2015), a characterization of the implicit bias in special separable cases (Gunasekar et al., 2018a;b), and a more fine-grained analysis for neural networks exploiting their specific structure (Allen-Zhu & Li, 2019; Arora et al., 2019). More recently, there has been a growing interest towards understanding the *interpolation regime* for overparameterized function fitting, where SGD is often the basic object of analysis (Belkin et al., 2019; Mei & Montanari, 2019).

Finally, empirical questions on the generalization of adaptive gradient methods were brought to the forefront by Wilson et al. (2017), who exhibit empirical and theoretical situations where adaptive methods generalize poorly. Building on this premise, Keskar & Socher (2017) suggest switching from Adam and SGD during training. Smith & Topin (2019) develop a doctrine of "superconvergence" which eschews adaptive methods. Reddi et al. (2018) point out some pathological settings where Adam fails to converge, and amends the algorithm accordingly. Schneider et al. (2019) note some sociological problems leading to misleading research on optimizer selection, providing a benchmarking suite for fairer hyperparameter searches, with mixed preliminary conclusions.

## 2  BACKGROUND

We begin by reviewing the stochastic optimization setting, and giving rigorous definitions of the adaptive gradient methods commonly used in practice.

## 2.1  OPTIMIZATION SETTING

We will focus on stochastic optimization tasks of the form

$$\text{minimize} \ \ F(w) \stackrel{\text{def}}{=} \mathbb{E}_{z\sim\mathcal{D}}[f(w; z)],$$

where the expectation is over a random variable $z$ whose distribution $\mathcal{D}$ is initially unknown; in machine learning, $z$ often represents a pair $(x, y)$ of an example $x$ and its corresponding label $y$, drawn

from an unknown population. A stochastic optimization algorithm is given a sample $z_1, \ldots, z_T \sim \mathcal{D}$ from the underlying distribution, and produces a point $\bar{w} \in \mathbb{R}^d$ whose population loss $F(\bar{w})$ is as close as possible to that of the minimizer $w^\star = \arg\min_w F(w)$. Often, iterative (first-order) optimization methods maintain a sequence of iterates $w_1, \ldots, w_T$ and, at each step $t$, use the *stochastic gradient*

$$g_t = \nabla f(w_t, z_t)$$

to form the next iterate $w_{t+1}$. The simplest stochastic optimization method is Stochastic Gradient Descent (SGD), whose update rule at step $t$ takes the form

$$w_{t+1} \leftarrow w_t - \eta_t\, g_t,$$

where $\eta_t > 0$ is a *step size* (or *learning rate*) parameter, whose scale and time-varying behavior are typically determined via hyperparameter search.

## 2.2 ADAPTIVE GRADIENT METHODS

Adaptive gradient methods comprise a general family of iterative optimization algorithms which attempt to automatically adapt to anisotropic gradient and parameter sizes. Often, an adaptive method will incorporate a different (adaptively computed) step size for each entry of the gradient vector. More specifically, the most common adaptive methods divide each parameter's gradient update by a second-moment-based estimate of the scale of its historical gradients. A concise way to unify this family of adaptive methods is given by following update equation (starting from an arbitrary initializer $w_0$):

$$w_{t+1} \leftarrow w_t - \alpha_t H_t^{-1} g_t + \beta_t H_t^{-1} H_{t-1}(w_t - w_{t-1}). \tag{1}$$

The above update expresses a broad family of methods including SGD, momentum (i.e., Polyak's heavy-ball method), AdaGrad, RMSprop, and Adam. The particular instantiations of the parameters $\alpha_k, \beta_k$ are summarized below:

|  | SGD | HB | AdaGrad | RMSprop | Adam |
|---|---|---|---|---|---|
| $G_t$ | $I$ | $I$ | $G_{t-1} + D_t$ | $\beta_2 G_{t-1} + (1-\beta_2)D_t$ | $\frac{\beta_2}{1-\beta_2^t}G_{t-1} + \frac{(1-\beta_2)}{1-\beta_2^t}D_t$ |
| $\alpha_k$ | $\alpha$ | $\alpha$ | $\alpha$ | $\alpha$ | $\alpha\frac{1-\beta_1}{1-\beta_1^t}$ |
| $\beta_k$ | $0$ | $\beta$ | $0$ | $0$ | $\frac{\beta_1(1-\beta_1^{t-1})}{1-\beta_1^t}$ |

Table 1: Parameter settings for common optimization algorithms in the unified framework of Equation 1. Here, $D_t \stackrel{\text{def}}{=} \text{diag}(g_t^{\odot 2})$ and $G_t \stackrel{\text{def}}{=} H_t^{\odot 2}$; here, $\odot 2$ denotes the entrywise square of a vector or matrix. We omit the $\epsilon$ parameters in the adaptive methods, see the discussion in Section 3.1.

# 3 PRACTICAL ASPECTS OF ADAPTIVE OPTIMIZATION

In this section, we compile some lesser-known practices in the usage of adaptive methods, which we have found to help consistently across large-scale experiments. We emphasize that this collection is restricted to simple ideas, which do not add extraneous hyperparameters or algorithmic alternatives.

## 3.1 THE $\epsilon$ HYPERPARAMETER

The general AdaGrad update, as originally proposed by Duchi et al. (2011), includes a parameter $\epsilon$ to allow for convenient inversions. Formally, the update looks like:[1]

$$w_{t+1} = w_t - \eta\, H_t^{-1} g_t; \quad H_t = \left[\sum_{k=1}^t \text{diag}(g_t^{\odot 2})\right]^{1/2} + \epsilon \cdot I$$

The inclusion of $\epsilon$ in the the original proposal of the above updates seems to have been made for convenience of presenting the theoretical results. However, in practice, this $\epsilon$ parameter often turns

---

[1]Adam handles the $\epsilon$ parameter slightly differently; see Kingma & Ba (2014).

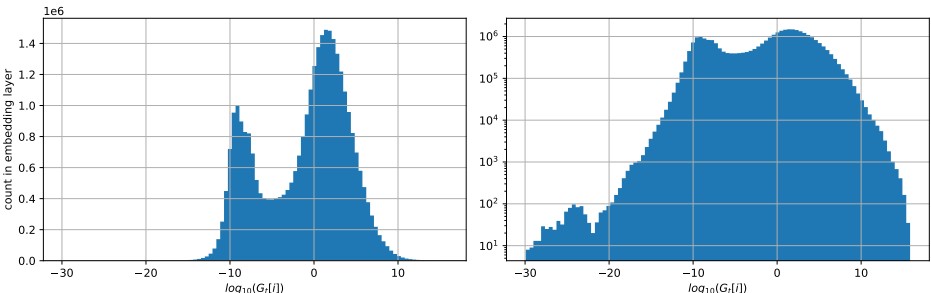

Figure 1: Distribution of nonzero accumulator values $G_t[i] = \sum_{\tau=1}^{t} g_\tau[i]^2$ in the embedding layer of the Transformer from Section 4.1, after $t = 116200$ training iterations. *Left:* Histogram of accumulator values, with counts on a linear scale. *Right:* Same plot as left, except with logarithmic vertical scale, showing prevalence of extremely small values.

out to be a parameter that should be tuned depending on the problem instance. The default value of this parameter in the standard implementations of the algorithm tend to be quite high; e.g., in Tensorflow (Abadi et al., 2016) it is $0.1$ which can be quite high. As large values would result in AdaGrad reducing to SGD with an implicit $1/\sqrt{\epsilon}$ learning rate, and losing out on all adaptive properties (RMSprop and Adam implementations also have an equivalent epsilon parameter). The effect can be seen in Figure 4 which shows that along many coordinates the accumulated second moments of the gradient are very small, even in the middle of training.

At least one work (Agarwal et al., 2019) remarks that the ability to choose a large $\varepsilon$ in a second-moment-based adaptive method might be a *feature* rather than a shortcoming; the smooth interpolation with SGD may improve the stability of more exotic optimizers. This does not appear to be the case for diagonal-matrix adaptive methods, in the NLP setting investigated in this paper.

Instead, we suggest removing this hyperparameter altogether and justify it in Section 4.2, and performing the AdaGrad update with the pseudoinverse instead of the full inverse. Then, the update is given by the following:

$$w_{t+1} = w_t - \eta \, H_t^\dagger g_t, \tag{2}$$

where $A^\dagger$ denotes the Moore-Penrose pseudoinverse of $A$ and with the preconditioning matrices updated as before. The above means that if there is a coordinate for which the gradient has been $0$ thus far we make no movement in that coordinate. This fix which can similarly be applied to the full matrix version of AdaGrad, does not affect the regret guarantees of AdaGrad. We provide an analysis in the Appendix B, verifying as a sanity check that the standard AdaGrad regret bounds continue to hold when $\varepsilon$ is completely removed.

## 3.2 MOMENTUM

A key distinction between AdaGrad, RMSprop and Adam is as follows: AdaGrad does not include momentum, and there is a per-parameter learning rate which is inverse of the accumulated gradient squares for that parameter. RMSprop as described in Hinton et al. (2012) uses exponential moving averaging rather than straightforward accumulation that AdaGrad relies on, and Adam modifies RMSprop to add momentum for the gradients along with a bias-correction factor. Note that implementation of RMSprop can vary based on the software library; e.g., TensorFlow (Abadi et al., 2016) includes modification to include momentum, while Keras API (Chollet et al. (2015)) does not. We note that it is straightforward to extend AdaGrad to incorporate heavy-ball momentum, where we start with $\bar{g}_0 = 0$ (and from a certain initialization $w_0$) and iteratively update:

$$\bar{g}_t \quad \leftarrow \quad \beta \bar{g}_{t-1} + (1-\beta) H_t^{-1} g_t;$$
$$w_{t+1} \leftarrow w_t - \eta \bar{g}_t.$$

## 3.3 LEARNING RATE WARMUP

The original definition of the Adam optimizer (Kingma & Ba, 2014) includes a *bias correction* term, in which the moment estimates are multiplied by the time-varying scalars $(1 - \beta_1^t)$ and $(1 - \beta_2^t)$. As

mentioned in the original paper, the bias correction can equivalently be written as an update to the learning rate. In the notation of Table 1:

$$\alpha_t = \alpha \cdot \frac{\sqrt{1 - \beta_2^t}}{1 - \beta_1^t}.$$

As can be seen from Figure 2, for the typical values of $\beta_1 = 0.9$ and $\beta_2 = 0.999$, the effective multiplier on the learning rate essentially resembles an external *warmup* applied on top of the learning rate. The general heuristic of including a warmup phase at the beginning of training has gained significant popularity in state-of-the-art empirical works; see, for example, Goyal et al. (2017); Shazeer & Stern (2018); Keskar et al. (2019); Radford et al. (2019); Devlin et al. (2018).

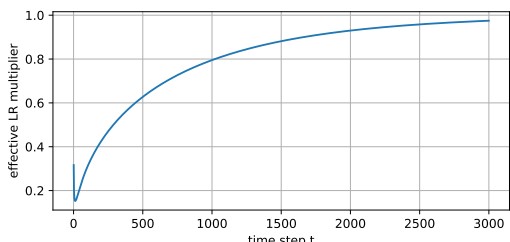

Figure 2: The effective learning rate multiplier of Adam as function of steps for the commonly found default hyperparameters $\beta_1 = 0.9, \beta_2 = 0.999$ in deep learning libraries.

Applying such a warm up externally on Adam results in now 3 hyper-parameters ($\beta_1, \beta_2$ and now the amount of warmup) conflating with each other, making hyper-parameter tuning difficult. Instead we suggest to *complete disable* this bias correction altogether and use an explicit warmup schedule in place of it. We use such a schedule in all of our experiments for SGD as well as adaptive optimizers as we find that it helps consistently across language modelling experiments.

One motivation for warmup during the initial stages of training is that for adaptive updates, the squared norm of the preconditioned gradient during the initial stage is quite large compared to the scale of the parameters. For the initial steps the preconditioned gradient squared norm is proportional to the number of coordinates with non-zero gradients where as the squared norm of the parameters are proportional to the number of nodes. Therefore adaptive methods are naturally forced to start with a smaller learning rate. The warmup in such a case helps the learning rate to rise up while the norm of the gradients fall sharply as training proceeds.

### 3.4 LEARNING RATE DECAY SCHEDULE

Learning rate decay schedules are one of the hardest to tune albeit a crucial hyperparameter of an optimizer. Stochastic gradient like algorithms, domain specific learning rate schedules have been derived over time with a lot of care and effort, examples include Resnet-50 on ImageNet-2012 where state of the art configuration of SGD+Momentum follows a stair-case learning rate schedule (while other type of schedules maybe possible). Adaptive algorithms apriori come with a potential promise of not requiring massive tuning of these schedules as they come with an in-built schedule with the caveat that AdaGrad variants like Adam or RMSprop does not enjoy a data-dependent decay like AdaGrad due to the presence of a non-zero decay factor and requires an external learning rate decay schedule. Even for experiments in Kingma & Ba (2014) which introduces Adam optimizer has experiments to include a $1/\sqrt{T}$ decay schedule for convergence. In our experiments, we found this implicit decay of AdaGrad to be sufficient for achieving superior performance on training a machine translation model, while an external decay rate was necessary for training the Resnet-50 on ImageNet-2012 to high accuracy.

### 4 EXPERIMENTS

We study the empirical performance of various optimization methods for training large state-of-the-art deep models, focusing on two domains: natural language processing (machine translation) and image recognition.

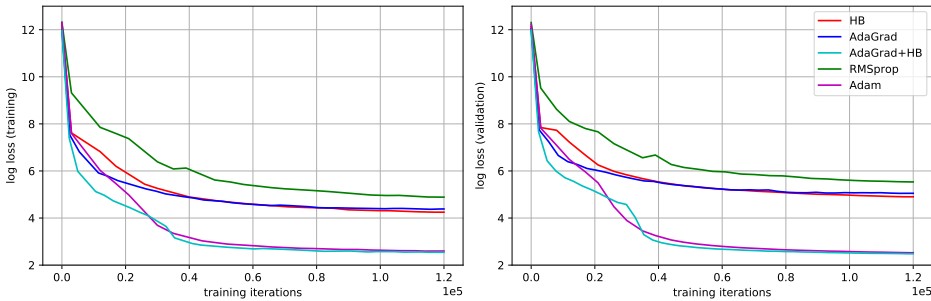

Figure 3: Training loss (i.e. log-perplexity) curves for several optimization methods on the `transformer-big` model on the WMT14 en→fr dataset.

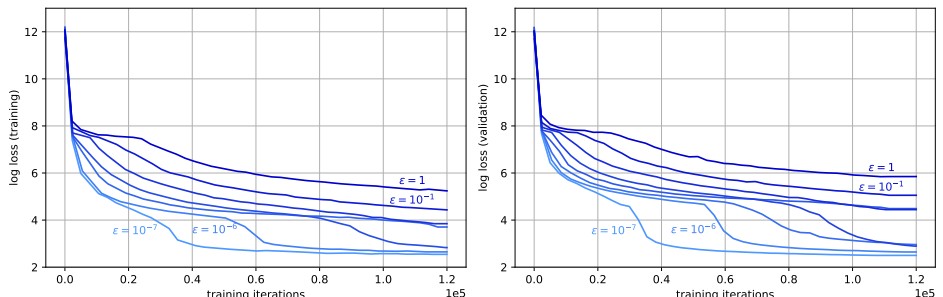

Figure 4: Convergence of AdaGrad with momentum in the translation setting of Section 4.1, varying only the initial accumulator value $\epsilon$. Note that using the default value $\epsilon = 10^{-1}$ in the TensorFlow AdaGrad implementation might cause dramatic misconceptions about poor convergence.

### 4.1 MACHINE TRANSLATION WITH TRANSFORMER

We study the convergence of various optimization methods when training a Transformer model (Vaswani et al., 2017) for machine translation. We used the larger Transformer-Big architecture (Chen et al., 2018); this architecture has 6 layers in the encoder and decoder, with 1024 model dimensions, 8192 hidden dimensions, and 16 attention heads. It was trained on the WMT'14 English to French dataset (henceforth "en→fr") that contains 36.3M sentence pairs. All experiments were carried out on 32 cores of a TPU-v3 Pod (Jouppi et al., 2017) and makes use of the Lingvo (Shen et al., 2019) sequence-to-sequence TensorFlow library.

We compared several optimization methods for training; the results are reported in Fig. 3. We see that a properly tuned AdaGrad (with $\varepsilon = 0$ and added momentum) outperforms Adam, while SGD with momentum, plain AdaGrad and RMSprop perform much worse on this task. These results illustrate that adaptivity and momentum are both extremely effective in training these models.

### 4.2 SENSITIVITY TO $\varepsilon$

In Section 3.1, we proposed an "$\varepsilon = 0$" variant of AdaGrad. Here we empirically motivate this modification, by investigating the effect of the parameter $\varepsilon$ on the performance of AdaGrad. We train the Transformer model from above on the en→fr dataset using AdaGrad while varying the value of $\varepsilon$. The results are given in Fig. 4. We see drastic improvement in convergence as we lower the value of $\varepsilon$ down to $10^{-7}$ (lower values do not improve convergence further and are thus omitted from the figure).

To see where these dramatic improvements come from, we also visualize in **??** the histogram of the square gradient values for the embedding layer of the model at step $t = 116200$, which indicates that a large fraction of the cumulative gradient entries have extremely small magnitudes. The choice of $\varepsilon$ is thus important, and justify our prescription of removing the dependency all-together instead of tuning it as a separate hyper-parameter.

### 4.3 IMAGE CLASSIFICATION WITH RESNET-50

Next, we trained a ResNet-50 architecture (He et al., 2015) on the Imagenet-2012 (Deng et al., 2009) dataset. The task is to classify images as belonging to one of the 1000 classes. Our training setup consists of 512 cores of a TPU v3 Pod and makes use of a relatively large batch size of 16386. As a baseline, we considered SGD with momentum with a highly-tuned staircase learning rate schedule, that achieves 75.3% test accuracy after 90 epochs. We compared several optimization methods on this task as seen in Fig. 5: the straightforward application of AdaGrad (with a fixed $\varepsilon$ and with heavy ball momentum) achieves only a paltry 63.94% test accuracy. Noticing that AdaGrad implicit decay schedule does not decay sufficiently fast, an external decay rate was added starting at epoch 50 of the form $(1 - \frac{\text{current epoch - 50}}{50})^2$. This change was sufficient for AdaGrad to reach a test accuracy of 74.76%—a drastic $>\sim 10\%$ jump. As demonstrated, learning rate schedule is a highly important hyperparameter and requires tuning for each task. E.g., the baseline SGD is highly tuned and follows an elaborate stair case learning rate to reach 75% test accuracy.

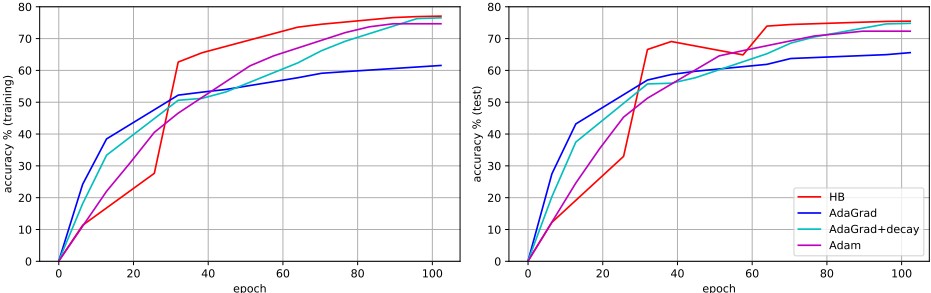

Figure 5: Training curves for top-1 training and test accuracy of a ResNet-50 on ImageNet, trained with several optimization methods. Without the cyan curve, the plots suggest poor generalization of adaptive methods, despite their head start in optimization. However, the gap is closed upon a careful tuning of AdaGrad with momentum.

### 4.4 REVISITING THE EXPERIMENTS OF WILSON ET AL. (2017)

We attempted to reproduce the experiments from Wilson et al. (2017), using the same codebases and identical hyperparameter settings. Although we were able to replicate some of their findings on these smaller-scale experiments, others appear to be sensitive to hyperparameter tuning, and perhaps subtle changes in the deep learning software and hardware stack that have occurred during the two years since the publication of that paper. In this section, we summarize these findings.

**Image classification.** On the classic benchmark task of CIFAR-10 classification with a VGG network (Simonyan & Zisserman, 2014), we were able to replicate the (Wilson et al., 2017) results perfectly, using the same codebase[2]. We repeated the hyperparameter search reported in the paper, found the same optimal base learning rates for each optimizer, and found the same stratification in performance between non-adaptive methods, Adam & RMSprop, and AdaGrad.

**Character-level language modeling.** Curiously, our replication of the language modeling experiment using the same popular repository[3] was successful in reproducing the optimal hyperparameter settings, but resulted in an opposite conclusion. Here, SGD found the objective with the smallest training loss, but Adam exhibited the best generalization performance. We believe that software version discrepancies (our setup: CUDA 10.1, cuDNN 7.5.1) may account for these small differences.

**Generative parsing.** We turn to the Penn Treebank (Marcus et al., 1994) constituency parsing code[4] accompanying (Choe & Charniak, 2016). Using the same architectural and training protocol modifications as specified in (Wilson et al., 2017), we were able to get the model to converge with

---

[2]https://github.com/szagoruyko/cifar.torch
[3]https://github.com/jcjohnson/torch-rnn
[4]https://github.com/cdg720/emnlp2016

each optimizer. However, for two of the settings (SGD and RMSprop), the best reported learning rates exhibited non-convergence (the fainter curves in Figure 6). Similarly as the above experiment, the ranking of optimizers' training and generalization performance differs from that seen in the original report.

Finally, Wilson et al. (2017) include a fourth set of experiments, generative parsing of Penn Treebank, using the code[5] accompanying (Cross & Huang, 2016). Unfortunately, this DyNet (Neubig et al., 2017) implementation, which was last updated in 2016, encountered a fatal memory leak when training with our DyNet 2.1 setup.

All relevant plots are given in Figure 6, with color codes selected to match Figures 1 and 2 in Wilson et al. (2017). Together, these experiments are further evidence for a widespread reproducibility crisis in deep learning: despite the authors' exceptional transparency in disclosing their optimizer selection and evaluation protocol, these benchmarks have turned out to be brittle for unknown reasons. Along the same lines as the random-seed-tuning experiments of Henderson et al. (2018), this suggests that there are further technical complications to the problems of credible optimizer evaluation addressed by Schneider et al. (2019), even on well-known supervised learning benchmarks.

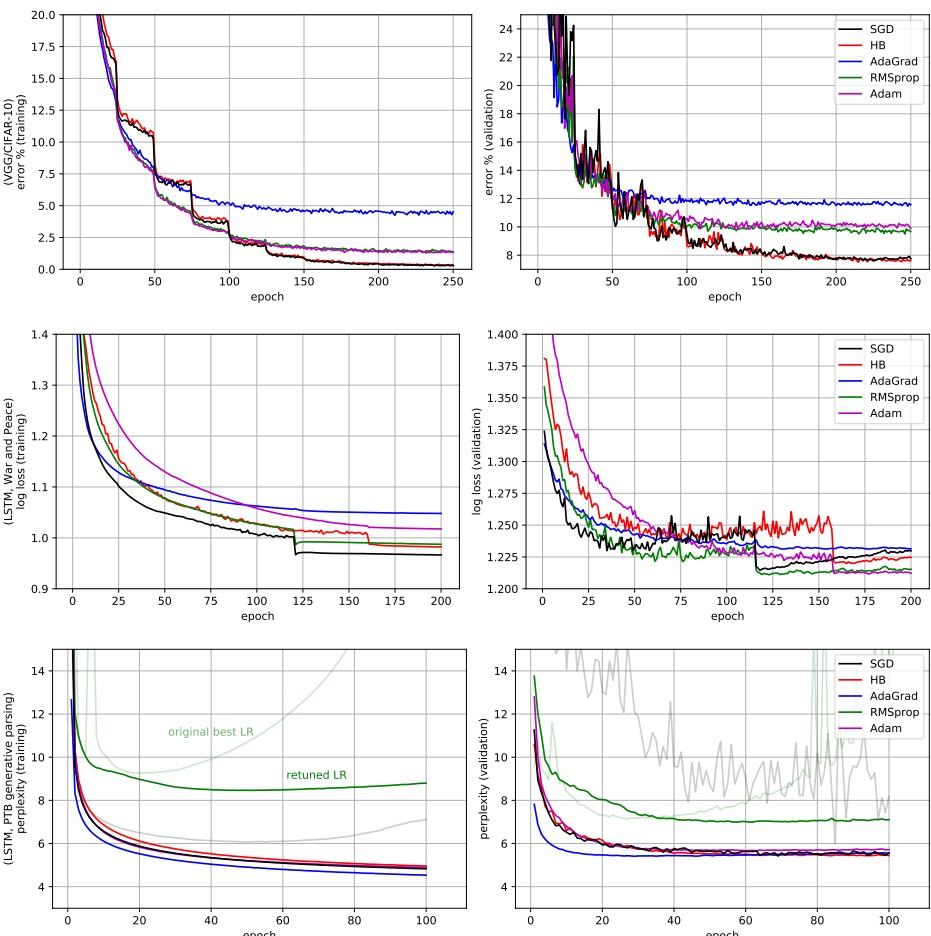

Figure 6: Attempted replication of the results of Wilson et al. (2017). *Top:* Image classification with a CNN. Results match perfectly, with non-adaptive methods generalizing the best. *Middle:* Character-level language modeling with a 2-layer LSTM. The original reported hyperparameters are the best and all optimizers converge to reasonable solutions, but contradictory conclusions about generalization arise. *Bottom:* 3-layer LSTM for generative parsing. Training does not converge with all reported learning rates; conclusions about generalization are unclear.

---

[5]https://github.com/jhcross/span-parser

## 5 THEORETICAL DISCUSSION

In this section we provide two simple examples of stochastic convex problems where it can be seen that when it comes to generalization both AdaGrad and SGD can be significantly better than the other depending on the instance. Our purpose to provide both the examples is to stress our point that the issue of understanding the generalization performance of SGD vs. adaptive methods is more nuanced than what simple examples might suggest and hence such examples should be treated as qualitative indicators more for the purpose of providing intuition. Indeed which algorithm will perform better on a given problem, depends on various properties of the precise instance.

### 5.1 EXAMPLE WHERE SGD > ADAGRAD

We provide a brief intuitive review of the construction provided by Wilson et al. (2017); for a precise description, see Section 3.3 of that paper. Consider a setting of overparameterized linear regression, where the true output (i.e. dependent variable) $y \in \{\pm 1\}$ is the first coordinate of the feature vector (independent variable) $x$. The next two coordinates of $x$ are "dummy" coordinates set to 1; then, the coordinates are arranged in blocks which only appear once per sample, taking the value of $y$.

The key idea is that in this setting, the solution space that AdaGrad explores is always in the subspace of the sign vector of $X^\top y$. As a result, AdaGrad treats the first three coordinates essentially indistinguishably putting equal mass on each. It can then be seen that for any new example the AdaGrad solution does not extract the true label information from the first three coordinates and hence gets the prediction wrong, leading to high generalization error; the other distinguishing features belong to the new unique block which are set to 0 for the AdaGrad solution, as it has not seen them.

### 5.1.1 EXAMPLE WHERE ADAGRAD > SGD

This example is motivated from the original AdaGrad paper (Duchi et al., 2011), adapted to the overparameterized setting. Consider a distribution $\mathcal{Z}$ supported over $\{0, 1\}^d$ with equal $1/d$ mass over vectors with exactly one 1 and 0 mass everywhere else. Let the label distribution be always $y = 1$. Consider sampling a dataset $S$ of size $c \cdot d$ where $c \leq 1$ (corresponding to the overparameterized setting) and consider the hinge loss

$$f_t(x) = [1 - y_t(z_t^\top x_t)]_+$$

where $(z_t, y_t)$ denotes the $t$-th (example, label) pair. Note that there is an optimal predictor given by the all-ones vector.

Running AdaGrad in such a setting, it can be seen that the first time a vector that has not appeared yet is sampled, AdaGrad quickly adapts by setting the coordinate corresponding to the vector to 1 and thereby making 0 error on the example. Therefore after one epoch of AdaGrad ($cd$ steps), the training error reduces to 0 and the average test error becomes roughly $(1 - c)$. On the other hand, for SGD (with an optimal $1/\sqrt{t}$ decay scheme) after say $cd/2$ steps, the learning rate reduces to at most $O(1/\sqrt{d})$ and therefore in the next $cd/2$ steps SGD reduces the error at most by a factor of $O(1 - 1\sqrt{d})$, leading to a total test error of at least $\sim (1 - c/2)$ after a total of $cd$ steps. This is significantly smaller than the error achieved by AdaGrad at this stage. Further note that to get down to the same test error as that achieved by AdaGrad, it can be seen that SGD requires at least $\Omega(\sqrt{d})$ times more steps than AdaGrad.

## 6 CONCLUSION

We have presented an empirical retrospective on adaptive gradient methods, along with several practical algorithmic insights, concluding that the adaptive gradient methods do not in general carry a generalization penalty. In no way do we purport to settle the debate about optimizer selection; at the very least, this would require a massive hyperparameter search, which would need to be updated as trends evolve and new architectures and benchmarks emerge. Instead, our experiments aim to challenge some possible misconceptions, while pointing out the simple algorithmic suggestions they imply. In doing so, we hope to contribute to a greater clarity on the messy frontier of optimizer selection and tuning.

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

## A  APPENDIX

### A.1  DETAILS OF EXPERIMENTS

#### A.1.1  TRANSFORMER-BIG ON WMT14 ENGLISH TO FRENCH

Models were trained with 384 batch size with a linear warm-up over 40k steps. We did a grid search over the parameters. Ranges of these were: Learning rate $\eta \in [10^{-6}, 10^0]$, Momentum $\beta_1 \in \{0.9, 0.99\}$, and eecond-moment coefficient for Adam/RMSprop $\beta_2 \in 0.99, 0.999$.

1. **AdaGrad:** Following versions of AdaGrad had $\epsilon$ fixed to zero. (Note we indeed add an $\epsilon$ of $10^{-30}$ such that $\frac{1}{\sqrt{\epsilon}}$ is not infinity.)

    (a) **Vanilla:** $\eta = 0.01$
    (b) **Heavy Ball Momentum:** $\beta_1 = 0.9, \eta = 0.05$

2. **Adam:** $\beta_1 = 0.9$, $\beta_2 = 0.999$, with maximal learning rate of 0.000075, and followed by decay as suggested in Vaswani et al. (2017).

3. **RMSprop:** $\beta_2 = 0.999$, $\eta = 0.000075$

4. **Momentum:** $\beta_1 = 0.9$, $\eta = 0.08$

### A.2  RESNET-50 ON IMAGENET-2012

Models were trained with 16384 batch size, with L2 regularization and label smoothing coefficients set as 1e-4 and 0.1 respectively.

1. **SGD+Momentum(HB):** follows a staircase learning rate schedule where learning rate is ramped up linearly from 0 to 6.4 over the first 5 epochs, followed by 10x drop in learning rate at 30 epochs, 60 epochs and 80 epochs. Momentum is set to 0.9.

2. **AdaGrad+Decay:** We linearly increase the learning rate from 0 to 0.1 over the first 50 epochs, followed by an external decay rate, that follows $(1 - \frac{\text{epoch since } 50}{50})^2$. AdaGrad here refers to the epsilon fixed version with heavy-ball momentum coefficient set to 0.9.

3. **Adam:** Decay rate follows the same schedule as AdaGrad decay, however, uses a maximal learning rate of 0.001.

4. **AdaGrad:** without decay linearly increases the learning rate from 0 to 0.1 over the first 20 epochs. AdaGrad here refers to the epsilon fixed version with heavy-ball momentum.

## B  ANALYSIS OF ADAGRAD WITHOUT $\epsilon$

In Section 3.1 we provide a method for setting the $\epsilon$ to 0. The general update is performed as follows

$$w_{t+1} = w_t - \eta \, H_t^\dagger g_t; \quad G_{t+1} = G_t + g_t g_t^\mathsf{T},$$

where the preconditioning matrix $H_t$ is updated in two possible ways:

$$\textbf{diagonal}: H_t \stackrel{\text{def}}{=} \text{diag}(G_t)^{1/2}; \qquad \textbf{full}: H_t \stackrel{\text{def}}{=} G_t^{1/2}.$$

The following theorem whose proof is very similar to the original analysis in Duchi et al. (2011) shows that the above modification leads to no change in the regret guarantee of AdaGrad.

**Theorem 1.** *The regret of the AdaGrad algorithm with updates implemented as* (2) *is bounded as*

$$\text{(full)} \quad Regret \ \le \ O\left(\max_{t \le T} \|w_t - w^*\|_2 \, \text{tr}(G_T^{1/2})\right)$$

$$\text{(diagonal)} \quad Regret \ \le \ O\left(\max_{t \le T} \|w_t - w^*\|_\infty \sum_{i=1}^d \sqrt{\sum_{t=1}^T g_{t,i}^2}\right)$$

We now provide a quick proof sketch for the above highlighting the main parts of the proof that change from the standard version. As in the original proof we consider the case of linear loss functions at every step given by $g_t$. The first step is to note that the following relationship holds directly by the definition of the updates.

$$g_t^\top (x_t - x^*) = \frac{1}{2\eta}(\|x_t - x^*\|_{H_t}^2 - \|x_{t+1} - x^*\|_{H_t}^2) + \frac{\eta}{2}(g_t^\top H_t^\dagger H_t H_t^\dagger g_t)^2.$$

The analysis now follows in the standard way by summing the above over time and analyzing the first and second summation separately. The first term is the same as the standard analysis and therefore leads to no change.

Further more for the second term the idea of the original proof is to show that

$$\sum_t (g_t^\top H_t^\dagger H_t H_t^\dagger g_t)^2 \leq 2\mathrm{tr}(H_T).$$

The above statement follows in the same way as the original proof with care for pseudoinverses. For instance in the diagonal version we can apply Lemma 4 from Duchi et al. (2011) along each coordinate separately applying the lemma from the first time the coordinate sees a non-zero gradient and ignoring everything before as it is 0.

