# OpenReview forum: "Revisiting the Generalization of Adaptive Gradient Methods"
_ICLR.cc/2020/Conference — Reject_

### Official Review · AnonReviewer2 · 2019-10-22
**Official Blind Review #2**

**Rating:** 3

**Review:**

This paper does an empirical study for the problem of "whether acceleration harms the empirical learning performance". Based on the study, the authors propose some empirical suggestions to fix the generalization gap for acceleration methods.

Overall, this paper is a lack of insights and looks more like an experimental report. The significance of the paper is not enough. Please see questions below

Q1. "We re-examine this belief both theoretically and experimentally". Where is the theoretical part?
- I wish the authors can offer some useful Lemmas or Theorems, but Section 5 only offers some discussions.

Q2. Since the paper wants to promote some empirical observations, please add STD to all curves, like,  Wilson et al. (2017). Some curves are too close (e.g., those Figure 6), it is hard to judge their statistic significance.

Q3. "we synthesize a user’s guide to adaptive optimizers".
- After reading the paper several times, I am still kind of confused. What is the "user’s guide"? Could the authors make some short summary?
- Since the paper mainly does empirical study, this question is important.

Q4. Why not SGD in Figure 3?

Q5. "As demonstrated, learning rate schedule is a highly important hyperparameter and requires tuning for each task".
- So, how to tune the learning rate?
- Is the learning rate more important than the choice of the optimizer? If so, the problem with the acceleration this paper investigated has little meaning.

Q6. Is it better to carry on hyperparameter optimization on factors that can contribute to the final performance of each algorithm?
- In this way, we can see the extreme performance of each optimizer, the performance comparison condition on best possible hyperparameter can make more sense than what has been done in this paper.
- It will be good if authors can check this paper "Neural Optimizer Search with Reinforcement Learning" and then re-design their methodology.

**Experience Assessment:**

I have published in this field for several years.

**Review Assessment: Checking Correctness Of Derivations And Theory:**

I assessed the sensibility of the derivations and theory.

**Review Assessment: Checking Correctness Of Experiments:**

I assessed the sensibility of the experiments.

**Review Assessment: Thoroughness In Paper Reading:**

I read the paper thoroughly.

---

> ### Author Response · Authors · 2019-11-15
> **Response to R2**
>
> Thanks for the thorough review and many good questions and suggestions.
>
> @Q1 (Theory): By “theoretically” we were referring to the discussion of theoretical examples, intended to continue the discussion from Section 3 of [Wilson et al. ‘17]. We will package the discussions of Section 5 into formal theorems.
>
> @Q2 (Standard deviations): We will revise to include error bars for completeness. Of course, this will not change the conclusions of the experiment.
>
> @Q3 (“User’s guide”): We did not mean to misstate or overstate the contribution, and agree that this was a poor choice of wording. By “user’s guide” we meant to refer to the collection of observations and suggestions of aspects of adaptive optimizers which need not be tuned in a first cut. This is not a comprehensive toolbox, but rather a way to pare down the hyperparameter search space, towards a minimal manifestation of adaptivity. As mentioned to R3, we will overhaul the writing to reflect this.
>
> @Q4 (SGD comparison): The HB (Heavy Ball) curve is SGD with momentum but not adaptivity, providing the comparison between adaptive and non-adaptive methods here. As a side note, SGD without momentum on this architecture performed significantly worse.
>
> @Q5 (Learning rate schedule): There is no empirical consensus on learning rate schedule tuning, and we are not intending in this work to enter that debate. We are also not concluding here that LR schedule is more important than optimizer choice. Our discussion of LR schedules is exclusively to make the following point: the discrepancies between different adaptive optimizers may be explained by their different implicit LR schedules. This is, again, aimed towards paring down the hyperparameter space.
>
> @Q6 (Relation to optimizer search): Again, we are intending in this work to revisit controversial beliefs about the generalization of adaptive methods, not provide new optimizer tricks or state-of-the-art results.

---

### Official Review · AnonReviewer3 · 2019-10-22
**Official Blind Review #3**

**Rating:** 3

**Review:**

This paper revisited the a common belief that adaptive gradient methods hurts generalization performances. The authors re-examine this in more depth and provide a new set of experiments in larger-scale, state-of-the-art settings. The authors claimed that with proper tuning, the performance of adaptive optimizers can mitigate the gap  with non-adaptive methods.

- It is great to have someone revisit and challenge the conventional ideas in the community. However, I did not find much insightful information in this paper. As the author mentioned, there are many recent works focusing on further improving the empirical generalization performances of adaptive gradient methods. The main aspects mentioned in the paper, \epsilon tuning, learning rate warmup and decaying schedule, are not something new and many of which are mentioned or used in the recent advances. This makes the contribution of this paper look like combining all the tricks together. The authors might want to carefully comment the differences with the recent advances

[1] Liu, Liyuan, et al. "On the variance of the adaptive learning rate and beyond." arXiv preprint arXiv:1908.03265 (2019).
[2] Loshchilov, Ilya, and Frank Hutter. "Decoupled weight decay regularization." ICLR 2019.
[3] Zaheer, Manzil, et al. "Adaptive methods for nonconvex optimization." Advances in Neural Information Processing Systems. 2018.
[4] Chen, Jinghui, and Quanquan Gu. "Closing the generalization gap of adaptive gradient methods in training deep neural networks." arXiv preprint arXiv:1806.06763 (2018).
[5] Luo, Liangchen, et al. "Adaptive gradient methods with dynamic bound of learning rate." arXiv preprint arXiv:1902.09843 (2019).

- In terms of tuning \epsilon, the authors mentioned that default setting in Tensorflow is 0.1 for Adagrad which is too high. However, most papers regarding adaptive gradient method usually set \epsilon as 10^-8. Pytorch set default value as 10^-10. In fact, Yogi paper mentioned above gives some different conclusions. In their experiments, they found that setting \epsilon to be a bit larger like 10^-3 give better results compared with 10^-8. I wonder if the authors examine the reasons for different conclusions here?

- at the end of page 6, missing reference for histogram


**Experience Assessment:**

I have published in this field for several years.

**Review Assessment: Checking Correctness Of Derivations And Theory:**

I carefully checked the derivations and theory.

**Review Assessment: Checking Correctness Of Experiments:**

I carefully checked the experiments.

**Review Assessment: Thoroughness In Paper Reading:**

I read the paper thoroughly.

---

> ### Author Response · Authors · 2019-11-15
> **Response to R3**
>
> Thanks for the review.
>
> @Insights: Our message is explicitly *not* to add new optimizer tricks. The main insight we wish to convey is that contrary to common belief, adaptive optimizers can generalize well. There are confounding factors (including bad choice of hyperparameters) that lead to the opposite misleading belief which we wish to disentangle in this paper (see next point). Thus, we have attempted to only cite the vast literature on recent advances in optimizer tricks when they include points important to our message.
>
> @Conclusions about \eps=0: We do not claim that there is no mileage to be obtained out of using a tuned positive \eps for different algorithms and models, especially since this has the interpretation of interpolating with SGD. The purpose of the \eps=0 suggestion, as well as all of our other algorithmic points, is to pare down the hyperparameter search space as much as possible, and obtain the simplest useful manifestation of adaptivity for which a consistent discussion of generalization can be made. Indeed as we demonstrate removing this epsilon hyperparameter by itself leads to a very effective and useful algorithm. We agree that this was not as clear as it should have been, and will overhaul the writing to reflect this intent.

---

### Official Review · AnonReviewer1 · 2019-10-24
**Official Blind Review #1**

**Rating:** 6

**Review:**

This paper revisits the question of whether adaptive methods deliver solutions with larger generalization errors than those of solutions found using SGD. The conclusion is that "which algorithm will perform better on a given problem, depends on various properties of the precise instance." They reach this conclusion by rerunning the same experiments with the same settings as in the 2017 Wilson et al. paper which showed empirical evidence that adaptive methods exhibit worse generalization error than SGD, and showing that the results have changed, due to extraneous factors like the fact that they run on different hardware. The only advantage of adaptive methods seems to be that the amount of tuning is less intensive (e.g., no need to handtune staircase decaying learning rates). They provide two mathematical examples to show that SGD (with a particular decay rate) and Agagrad can beat the other in terms of generalization error, depending on the setting.

The takeaway of this paper seems to be that adaptive methods and non-adaptive methods both need careful parameter tuning. They also suggest using Adagrad without adding a multiple of the identity to the approximate Hessian, and point out that externally imposed decay rates can help adaptive methods. I think the paper is well-written, and provides convincing evidence that neither SGD or adaptive methods dominates the other. I lean towards accept.

Comments:
Figure 5 doesn't show a significant gap between the test and training error for the adaptive methods, as claimed in its caption.
There is a missing figure reference at bottom of page 6

**Experience Assessment:**

I do not know much about this area.

**Review Assessment: Checking Correctness Of Derivations And Theory:**

I assessed the sensibility of the derivations and theory.

**Review Assessment: Checking Correctness Of Experiments:**

I assessed the sensibility of the experiments.

**Review Assessment: Thoroughness In Paper Reading:**

I read the paper thoroughly.

---

> ### Author Response · Authors · 2019-11-15
> **Response to R1**
>
> Thanks for the review.
>
> @Figure 5: Yes, what we mean is that AdaGrad by default does not perform as well as the others in terms of the achieved test accuracy. We will make the correction.
>
> @Missing reference: We will correct this in the paper.

---

### Decision · Program_Chairs · 2019-12-19

**Decision:**

Reject

**Comment:**

The paper combines several recent optimizer tricks to provide empirical evidence that goes against the common belief that adaptive methods result in larger generalization errors. The contribution of this paper is rather small: no new strategies are introduced and no new theory is presented. The paper makes a good workshop paper, but does not meet the bar for publication at ICLR.